Development of a cost-effective metabarcoding strategy for analysis of the marine phytoplankton community

Yoon Tae-Ho 1
Kang Hye-Eun 2
Kang Chang-Keun 3
Lee Sang Heon 4
Ahn Do-Hwan 5
Park Hyun hpark@kopri.re.kr 6
Kim Hyun-Woo kimhw@pknu.ac.kr 1 2
1 Interdiciplinary Program of Biomedical Engineering, Pukyong National University , Busan , Republic of Korea
2 Department of Marine Biology, Pukyong National University , Busan , South Korea
3 School of Earth Science & Environmental Engineering, Gwangju Institute of Science and Technology , Gwangju , Republic of Korea
4 Department of Oceanography, Pusan National University , Busan , South Korea
5 Division of Polar Life Sciences, Korea Polar Research Institute , Incheon , South Korea
6 Korea Polar Research Institute, Korea Ocean Research and Development Institute , Incheon , Republic of Korea
Woyke Tanja
Electronic publication date: 2016 Jun 15
Publication date: 2016
Volume: 4
Electronic Location ID: e2115
Received 2016 Feb 20; Accepted 2016 May 17
Copyright: ©2016 Yoon et al.
Copyright year: 2016
Copyright holder: Yoon et al.
License: This is an open access article distributed under the terms of the Creative Commons Attribution License, which permits unrestricted use, distribution, reproduction and adaptation in any medium and for any purpose provided that it is properly attributed. For attribution, the original author(s), title, publication source (PeerJ) and either DOI or URL of the article must be cited.
License URL: https://creativecommons.org/licenses/by/4.0/

Keywords: Barcode, Phytoplankton, Next-generation sequencing, Oceanography, Genomics

Funding: Ministry of Oceans and Fisheries, Korea This research was part of the project titled “Long-term change of structure and function in marine ecosystems of Korea,” funded by the Ministry of Oceans and Fisheries, Korea. The funders had no role in study design, data collection and analysis, decision to publish, or preparation of the manuscript.

==============================
We developed a cost-effective metabarcoding strategy to analyze phytoplankton community structure using the Illumina MiSeq system. The amplicons (404–411 bp) obtained by end-pairing of two reads were sufficiently long to distinguish algal species and provided barcode data equivalent to those generated with the Roche 454 system, but at less than 1/20th of the cost. The original universal primer sequences targeting the 23S rDNA region and the PCR strategy were both modified, and this resulted in higher numbers of eukaryotic algal sequences by excluding non-photosynthetic proteobacterial sequences supporting effectiveness of this strategy. The novel strategy was used to analyze the phytoplankton community structure of six water samples from the East/Japan Sea: surface and 50 m depths at coastal and open-sea sites, with collections in May and July 2014. In total, 345 operational taxonomic units (OTUs) were identified, which covered most of the prokaryotic and eukaryotic algal phyla, including Dinophyta, Rhodophyta, Ochrophyta, Chlorophyta, Streptophyta, Cryptophyta, Haptophyta, and Cyanophyta. This highlights the importance of plastid 23S primers, which perform better than the currently used 16S primers for phytoplankton community surveys. The findings also revealed that more efforts should be made to update 23S rDNA sequences as well as those of 16S in the databases. Analysis of algal proportions in the six samples showed that community structure differed depending on location, depth and season. Across the six samples evaluated, the numbers of OTUs in each phylum were similar but their relative proportions varied. This novel strategy would allow laboratories to analyze large numbers of samples at reasonable expense, whereas this has not been possible to date due to cost and time. In addition, we expect that this strategy will generate a large amount of novel data that could potentially change established methods and tools that are currently used in the realms of oceanography and marine ecology.

Introduction

Traditional phytoplankton surveys have relied mainly on microscopic observations, which require significant time and trained experts to obtain accurate data. This has created a bottleneck and is a major consideration in research planning. For these reasons, there is a need for a faster and more reliable standard method of performing detailed analysis of phytoplankton community structures. Alternatively, phytoplankton communities can be roughly analyzed based on different pigments using flow-cytometry (Dubelaar et al., 2007), which can sometimes be integrated with digital imaging systems (Campbell et al., 2010; Kachel & Wietzorrek, 2000; Olson, Shalapyonok & Sosik, 2003). However, such strategies aimed at replacing microscopic observations are still far from reliable for discriminating the numerous phytoplankton taxa.

To improve results from phytoplankton community structure analyses, molecular techniques are now being applied. The most widely used DNA markers include two protein-coding genes, protein D1 of photosystem-II reaction center (psbA) and a large subunit of the ribulose-1,5-diphophate carxoylase/oxygenase (rbcL) or plastid 16S rDNA region (Kirkham et al., 2013; Man-Aharonovich et al., 2010; McDonald et al., 2007; Paul, Alfreider & Wawrik, 2000; Zeidner et al., 2003). In particular, the 16S rDNA sequence has been well documented in the GenBank database (http://www.ncbi.nlm.nih.gov/) and more than 6,490 phytoplanktonic 16S sequences are currently deposited in the PhytoREF database (http://phytoref.org). Although there is still debate, universal primers targeting 16S rDNA are now most widely used (Decelle et al., 2015; Herlemann et al., 2011). However, it has been difficult to design a universal primer set that will amplify all phytoplankton taxa from cyanobacteria to eukaryotic algae in the 16S rDNA region, and most studies have analyzed specific taxonomic groups, especially for the bacterial communities (Asudi et al., 2016; Cruaud et al., 2014; Kitamura et al., 2016; Le Bescot et al., 2015; Logares et al., 2014; Massana et al., 2015; Valenzuela-González et al., 2016; Vierheilig et al., 2015). Although the sequence information from the 23S rDNA region is only a subset of that for 16S rDNA in the database, this region is considered an important marker for phytoplankton community structure when designing better universal primer sets that will cover most phytoplankton taxa (Folmer et al., 1994; Sherwood & Presting, 2007).

The rapid pace of technological advancement in DNA sequencing and bioinformatics analysis has meant that researchers can now analyze phytoplankton community structure simply by massive sequencing of DNA barcodes that are amplified from environmental samples, a process known as “metabarcoding.” In fact, several trials have already been conducted to analyze algal community structure using next generation sequencing (NGS) technology and generating large amounts of data for large survey areas at low cost, as compared with the Sanger method which uses dideoxynucleotide analogs (Eiler et al., 2013; Steven, McCann & Ward, 2012). These studies were dependent on the Roche 454 platform because its maximum read length (approximately 700 bp) is sufficient to identify to the species level. However, the cost for sequencing with the Roche 454 system remains high, and the Illumina platform has been introduced to replace this system and produce the same amount of sequence information at less than 1/100th of the cost (Luo et al., 2012). In fact, virtually all studies of microbial community structure are now being conducted using the Illumina sequencer (Caporaso et al., 2012; Degnan & Ochman, 2012; Jiang et al., 2013; Logares et al., 2014; Schmidt et al., 2013; Shakya et al., 2013; Staley et al., 2013) Still, phytoplankton community structure analysis using the Illumina system has yet to be widely applied, mainly due to the lack of universal primers that are suitable for the Illumina sequencer.

In this study, we developed a novel strategy of targeting the 23S rDNA region for phytoplankton community analysis using the MiSeq system (Illumina, San Diego, CA, USA). Performing large reads (404–411 bp) by paired-end sequencing (2X 300 bp) enabled us to analyze phytoplankton community structure at reduced cost (less than 1/20th of the cost with the Roche 454 system) and with high resolution. Our modified “universal primers” targeting the 23S rDNA region achieved a high level of coverage, amplifying most taxa of prokaryotic and eukaryotic phytoplankton while excluding the non-photosynthetic proteobacterial sequences. The reliability of this NGS strategy was confirmed by analyzing phytoplankton samples from different locations, depths, and seasons in the East/Japan Sea.

Materials and Methods

Sample collection and DNA extraction

Sampling was carried out in May and July 2014 under permitting (Department of Fisheries Resource Management-1728 & 2506) for the project “Long-term change of structure and function in marine ecosystems of Korea,” funded by the Ministry of Oceans and Fisheries, Korea. Field water samples from the surface (0 m) and at 50 m depth were collected using 10 L Niskin bottles at two sampling stations in the southwestern East/Japan Sea, Korea. Among various sample site options, station I (StI) was chosen as representative coastal water, whereas station II (StII) was in open sea. Samples were labeled with collection date, location, and depth, as follows: MIS: surface water from station I in May 2014 Station I surface water; MIIS: surface water from station II in May 2014; MIID: 50 m water from station 2 in May 2014; JIS: surface water from station I in July 2014; JIIS: surface water from station II in July 2014.

Water temperature and salinity were measured using an SBE-43 unit (Sea-Bird Electronics, USA). One liter of water was collected from each sampling site and filtered using a 0.45-µm GH Polypro membrane filter (Pall Corporation, New York, NY, USA). Chlorophyll-a concentration was measured using a Turner 10AU fluorometer (Turner Design Co., San Jose, CA, USA) as described by Parson, Maita & Lalli (1984). The filtered sample was ground into a fine powder using a mortar and pestle with liquid nitrogen. The total DNA from the homogenized phytoplankton was extracted using a DNeasy ® plant mini kit (Qiagen, Hilden, Germany) following the manufacturer’s instructions. Extracted DNA was quantified and qualified using a NanoDrop spectrophotometer ND-1000 (Thermo Scientific, Waltham, MA, USA) and was stored at −70°C until it was used.

Library preparation and sequencing

The “universal” primers for the mini-barcode were redesigned to target the plastid 23S rDNA region (Sherwood & Presting, 2007). According to multiple sequence alignments of 1,683 23S rDNA sequences (545 from eukaryotes and 1,138 from prokaryotes), several variations were detected in the forward primer. Based on these findings, the forward primer to be used were modified to increase specificity for phytoplankton taxa (Table 1). To lower the PCR-based bias, a nested PCR strategy with low cycle numbers was adopted. Two forward primers (A23SrVF1:GGACARAAAGACCCTATG and A23SrVF2: CARAAAGACCCTATGMAGCT) and two reverse primers (A23SrVR1:AGATCAGCCTGT TATCC and A23SrVR2: TCAGCCTGTTATCCCTAG) were designed and synthesized (Macrogen, Republic of Korea). The first and second PCR amplifications were performed under the following cycling conditions: initial denaturation at 94°C for 3 min, followed by 15 cycles at 94°C for 30 s, 55°C for 30 s, and 72°C for 30 s, with a final extension at 72°C for 3 min. The first PCR reaction mixture (40 µL) contained 10 ng of template, 2 µL of each primer (20 pmol), 4 µL of dNTPs (10 mM), 0.4 µL Ex Taq Hot Start Version (2 U) (Takara Bio Inc., Shiga, Japan), and 4 µL 10X buffer. The first amplicon was purified using the AccuPrep®Gel Purification Kit (Bioneer, Republic of Korea) and eluted with 20 µL elution buffer. The second PCR reaction mixture was the same as the first except that 2 µL of purified PCR product from the first PCR amplification was used. As with the first round, 2 µL of each primer was used for the second PCR (20 pmol). The products were separated by 1.5% agarose gel electrophoresis and stained with loading star (Dynebio, Sungnam, Republic of Korea). Amplicons of the expected sizes (approximately 410 bp) were purified using the AccuPrep®Gel Purification Kit (Bioneer, Daejeon, Republic of Korea). A library was constructed from the NGS data using the TruSeq®Sample Preparation kit V2 (Illumina, USA). The quality and quantity of the library were measured using a 2100 Bioanalyzer (Agilent Technologies, Santa Clara, CA, USA) and sequencing was performed using Illumina MiSeq (2 X 300 bp pair-ends) (Illumina, USA).

Table 1 Consensus Nucleotide sequences of forward primer targeting 23S rDNA region.

A23SrVF1		5′	G	G	A	C	A	R	A	A	A	G	A	C	C	C	T	A	T	G						3′	
A23SrVR1						C	A	R	A	A	A	G	A	C	C	C	T	A	T	G	M	A	G	C	T		
P23Rv˙f1 (Sherwood & Presting, 2007)		5′	G	G	A	C	A	G	A	A	A	G	A	C	C	C	T	A	T	G	A	A	–	–	–	3′	
Concensus sequence (Algae)			G	G	A	C	A	R	A	A	A	G	R	C	C	Y	Y	A	T	G	M	A	S	C	T		
Phylum	Total																										
Chlorophyta	113		G	G	A	C	A	A	A	A	A	G	A	C	C	C	T	A	T	G	A	A	G	C	T		
		113	113	113	113	113	113	113	113	113	113	113	113	113	113	113	113	113	113	113	113	113	113	113		
Cryptophyta	54		G	G	A	C	A	G	A	A	A	G	A	C	C	C	T	A	T	G	A	A	G	C	T		
		54	54	54	54	54	54	54	54	54	54	54	54	54	54	54	54	54	54	54	54	54	54	54		
Euglenophyta	141		G	G	A	C	A	G	A	A	A	G	A	C	C	Y (C/T)	T	A	T	G	M (A/C)	A	G	C	T		
		141	141	141	141	141	141	141	141	141	141	141	141	141	140/1	141	141	141	141	137/4	141	141	141	141		
Ochrophyta	26		G	G	A	C	A	G	A	A	A	G	A	C	C	C	T	A	T	G	A	A	G	C	T		
		26	26	26	26	26	26	26	26	26	26	26	26	26	26	26	26	26	26	26	26	26	26	26		
Streptophyta	198		G	G	A	C	A	G	A	A	A	G	A	C	C	C	T	A	T	G	A	A	G	C	T		
		198	198	198	198	198	198	198	198	198	198	198	198	198	198	198	198	198	198	198	198	198	198	198		
Dinophyta	13		G	G	A	C	A	R (A/G)	A	A	A	G	R (A/G)	C	C	C	T	A	T	G	A	A	S	C	T		
		13	13	13	13	13	1/12	13	13	13	13	12/1	13	13	13	13	13	13	13	13	13	12/1	13	13		
Cyanobacteria	195		G	G	A	C	A	G	A	A	A	G	A	C	C	C	Y (C/T)	A	T	G	A	A	G	C	T		
		195	195	195	195	195	195	195	195	195	195	195	195	195	195	194/1	195	195	195	195	195	195	195	195		
Proteobacteria-α	198		*	G	A	C	R (A/G)	G	A	A	A	R (A/G)	A	C	C	C	Y (C/T)	R	T	G	M (A/C)	A	C	C	T		
			198	198	198	1/197	198	198	198	198	2/196	198	198	198	198	96/102	109/89	198	198	98/100	198	198	198	198		
Proteobacteria-β	198		*	G	A	C	G	G	A	A	A	G	A	C	C	C	T	A	T	G	A	A	C	C	T		
			198	198	198	198	198	198	198	198	198	198	198	198	198	198	198	198	198	198	198	198	198	198		
Proteobacteria-γ	198		*	G	A	C	G	G	A	A	A	G	A	C	C	C	T	R	T	G	M (A/C)	A	C	C	T		
			198	198	198	198	198	198	198	198	198	198	198	198	198	198	1/197	198	198	181/17	198	198	198	198		
Proteobacteria-δ	151		*	G	R (A/G)	M (C/A)	R (A/G)	G	R (A/G)	A	A	R (A/G)	R	M (A/C)	C	C	Y (C/T)	N (A/G/C/T)	K (G/T)	D (A/G/T)	V (A/G/C)	W (A/T)	M (A/C)	Y (C/T)	Y (C/T)		
			151	147/4	147/4	21/126	151	147/4	151	151	4/147	147/4	4/147	151	151	32/119	15/132/2/2	9/142	5/142/4	90/7/54	143/8	4/147	146/5	4/147		
Proteobacteria-ε	198		*	G	A	G	G	G	R (A/G)	R (A/G)	A	G	A	C	C	C	T	G	T	G	S (C/G)	A	C	C	T		
			198	198	198	198	198	196/2	190/8	198	198	198	198	198	198	198	198	198	198	150/48	198	198	198	198		

Data processing of phytoplankton sequence data

Adapter/index and QV < 20 sequences from the raw data obtained were trimmed, and read length of <120 nucleotides were filtered using CLC Genomics Workbench v. 8.0 (CLC Bio, USA). End-paired amplicons were constructed using Mothur software v. 1.35.0 (Schloss et al., 2009) with the above 6 bp overlapping sequences and omitting any mismatches. The primer sequences were trimmed using the pdiffs = 0 option and Mothur software v 1.35.0. The PHYTOGEN database was constructed and updated as shown in Fig. 1. First, 3,581 sequences from the GenBank database (http://www.ncbi.nlm.nih.gov/) showing high similarity (>90%) to the amplicons obtained were included in the database. ``Uncultured'' sequences that had been deposited in GenBank by barcoding or metabarcoding projects were excluded. If the end-paired amplicons matched 100% with sequences in the PHYTOGEN database after alignment using BLASTn, the amplicons were named. If they did not match, the amplicons were named as Class name + PKNUE + numbers and entered into the PHYTOGEN database. To avoid errors, operational taxonomic units (OTUs) were not assigned for amplicons that were present in <0.1% of the community. Once the PHYTOGEN database was constructed, the total amplicons were analyzed by a BLASTn search and those with <99% identity in the PHYTOGEN database were described as ``others.''

Figure 1 Workflow of PHYTOGEN database process.

Results

Evaluation of the modified algal plastid 23S rDNA universal primers

As noted, the first step of the study was to modify the original “universal” primers targeting the plastid 23S rDNA region such that they excluded the non-photosynthetic proteobacteria (Table 1). Amplicons obtained from between the original universal primers and between the modified primers were compared (Table 2). In total, 185,773 (63 OTUs) and 625,917 (106 OTUs) amplicons were obtained using the original and modified primers, respectively (Table 2). Thirty-three OTUs were commonly identified by both primers. Thirty OTUs were exclusively identified from the sequence data generated with original universal primers, and all were proteobacteria. Seventy-three OTUs were exclusively identified from the sequence data generated with the modified primers, and all were eukaryotic algae. The difference in OTU numbers indicated that the taxonomic proportions of amplicons differed between the two sets of sequence data. Regarding the amplicons generated using the original universal primers, more than 84% were either unknown (43.9%) or proteobacteria (41.06%) according to the sequence data (Table 2). In contrast, regarding the amplicons generated using the modified primers, less than 24% were unknown (19.52%) or proteobacteria (4.55%), which reflected the presence of more algal amplicons and the elimination of non-photosynthetic proteobacteria (Table 2). The proportions of eukaryotic algal sequences belonging to the Haptophyta and Dinophyta were also greater when the modified primers were used. Collectively, the original universal primers were improved by the modifications that were made to the sequences.

Table 2 Comparison of phytoplankton sequences between the modified primer and original universal primer.

Phylum	Modified Primers	Original Primers	
	Amplicons	%	OTUs	Amplicons	%	OTUs	
Ochrophyta	398,588	63.68	76	24,006	12.92	31	
Unknown	122,178	19.52	−	81,562	43.90	−	
Haptophyta	47,816	7.64	22	2,198	1.18	6	
Proteobacteria	28,499	4.55	2	76,271	41.06	22	
Dinophyta	28,008	4.47	5	1,510	0.81	3	
Chlorophyta	828	0.13	1	226	0.12	1	
Total	625,917	100	106	185,773	100	63	

Construction of the algal ribosomal large subunit metabarcode database

To determine whether the modified 23S primers and PCR conditions could be directly applied to the phytoplankton survey, the described six samples were chosen and analyzed according to the different locations, seasons, and depths of the East/Japan Sea. After processing the raw reads, the number of end-paired amplicons obtained from each sample ranged from 290,082 to 639,836, and averaged 438,896 (Table 2). After several data processing steps, 345 OTUs were obtained from six samples combined (Table 2). Despite relatively small numbers of OTUs, only 95 (27.5%) matched the sequences previously listed in the GenBank database, indicating much lower numbers of the plastid 23S rDNA sequences compared with those of 16S rDNA in the database. In addition to the 95 OTUs identified, the remaining 250 OTUs were added to the PHYTOGEN database after assigning taxonomic ranks (phyla and classes) based on sequence similarity.

Figure 2 Composition of phytoplankton by the different cut-off similarity of OTUs.

(A) Each bar shows the ratio of phytoplankton phyla according to cut-off similarity from 90% to 99%. (B) Each bar shows the ratio of phytoplankton classes according to cut-off similarity from 90% to 99%.

Figure 3 Coverage of obtained OTUs in algal phyla.

Phylogenetic tree was constructed by the Neigbhor-joining (NJ) algorithm using Molecular Evolutionary Genetics Analysis (MEGA ver 6.0). The evolutionary distances were computed using the Kimura 2-parameter method.

To determine the cut-off value for phylum similarity, the OTUs obtained were screened from 90% to 99% (Fig. 2). The data revealed minimal differences in phyla between 90% and 97%, whereas the presence of significant unclassified OTUs increased as the cut-off was shifted 97–99% (Fig. 2). Considering this, the optimal cut-off for similarity was set at 97% for assigning the phylum of the OTUs. This indicated that the modified primers successfully amplified most phyla of the eurkayotic and prokaryotic algae in the samples, including the Dinophyta, Rhodophyta, Ochrophyta, Chlorophyta, Streptophyta, Cryptophyta, Haptophyta, and Cyanophyta (Fig. 3). Phyla not found in this study, such as Bigyra, Ciliophora, Katablepharidophyta, and Telonemia, either had ambiguous taxonomies or were rare at our sample sites. Sequences that had <97% identity with the PYTOGEN database were classified as “unknown,” and the proportions in this category ranged from 2.34% to 17.03% depending on the sample (Table 3). To ascertain whether further analysis of the OTUs at class level was possible, similarity from 97% to 100% was tested (Fig. 2B). While there was considerable difference from 98% to 100%, no significant difference was identified below 98%, indicating that 98% was the optimal similarity cut-off point for assigning class. We were unable to assign family level or below mainly due to the relatively small numbers of plastid 23S sequences in the database.

Table 3 Summary of end-paired phytoplanktonic contigs and processed OTUs in samples of East/Japan Sea.

Phylum	M I S	M II S	M II D	J I S	J II S	J II D	
	Contigs	Ratio (%)	OTUs	Contigs	Ratio (%)	OTUs	Contigs	Ratio (%)	OTUs	Contigs	Ratio (%)	OTUs	Contigs	Ratio (%)	OTUs	Contigs	Ratio (%)	OTUs	
Chlorophyta	4,815	0.75	21	7,261	2.30	22	8,656	2.26	15	7,036	1.13	13	98,420	23.91	19	8,279	2.85	14	
Cryptophyta	1,063	0.17	6	1,151	0.36	5	3,684	0.96	4	24	0.00	4	2,839	0.69	4	1,878	0.65	5	
Cyanobacteria	209	0.03	17	67,933	21.54	30	2,029	0.53	19	2,561	0.41	15	74,420	18.08	31	5,264	1.81	26	
Dinophyta	24,441	3.82	19	14,173	4.49	16	8,919	2.33	15	35,103	5.65	14	24,917	6.05	16	37,744	13.01	17	
Firmicutes	−	0.00	0	2	0.00	2	−	0.00	0	−	0.00	0	38	0.01	2	123	0.04	2	
Haptophyta	24,094	3.77	74	57,385	18.20	75	46,724	12.22	73	31,230	5.03	70	98,841	24.01	76	22,534	7.77	75	
Ochrophyta	552,850	86.40	118	115,611	36.66	119	67,856	17.74	113	510,993	82.31	109	30,117	7.32	118	82,463	28.43	123	
Unclassified	30,246	4.73	N/A	22,399	7.10	N/A	65,121	17.03	N/A	14,556	2.34	N/A	30,478	7.40	N/A	19,497	6.72	N/A	
Proteobacteria	1,177	0.18	19	27,885	8.84	46	178,140	46.57	41	17,715	2.85	32	50,790	12.34	46	111,346	38.38	54	
Rhodophyta	772	0.12	1	315	0.10	1	16	0.00	1	2	0.00	1	22	0.01	1	28	0.01	1	
Streptophyta	52	0.01	4	424	0.13	4	1,103	0.29	8	422	0.07	6	404	0.10	9	7	0.00	3	
Verrucomicrobia	117	0.02	2	807	0.26	2	253	0.07	2	1,200	0.19	2	312	0.08	3	923	0.32	4	
Plant	−	0.00	0	1	0.00	1	−	0.00	0	−	0.00	0	−	0.00	0	−	0.00	0	
Total	639,836	100	281	315,347	100	323	382,501	100	291	620,842	100	266	411,598	100	325	290,086	100	324	

Figure 4 (A) Ratio of algal phyla in each sample with different locations, depths, and seasons Each bar shows the ratio of phytoplankton phyla according to 97% cut-off similarity (B) Venn diagram of OTUs for the four surface samples in East/Japan Sea The venn diagram was generated with Draw Venn Diagram (http://bioinformatics.psb.ugent.be/webtools/Venn/).

Community structure analysis for phytoplankton of the East/Japan Sea

More OTUs were present in the samples collected from the open sea than in those collected from the coastal seas, and this was mainly to the prokaryotic population, including cyanobacteria and proteobacteria (Table 3). The costal surface water sample in July (JIS) contained the lowest number of OTUs (266), whereas the open surface water sample in July (JIIS) contained the highest number of OTUs (325) (Table 3). The numbers of exclusively identified OTUs at each sample site were very low, ranging from 10 to 24 (approximately 5% of the total OTUs [Fig. 4B]). The numbers of OTUs in each phylum were similar in each sample, and only their relative proportions varied (Table 3). For example, the numbers of OTUs belonging to the Ochrophyta were similar (109–123) among the six samples, but the corresponding abundance proportion varied from 7.32% to 86.40%. Species belonging to the Ochrophyta comprised >80% of those identified in the coastal water samples collected in May and July (Table 3), and most belonged to class Coscinodiscophyceae (Table 4). The surface and deep water samples from the open sea exhibited very different populations of cyanobacteria and proteobacteria (Table 3). Cyanobacteria occupied approximately 20% of the phytoplankton community in the surface water of the open sea samples from May and July, whereas the corresponding proportions in the coastal water samples were both 1%. Although the proportions of algal communities in the coastal surface waters and those in the deep waters of the open sea were similar at the different sampling times, a considerable difference was detected in the algal communities of the surface water of the open sea between May and July (Table 3). The proportion of species belonging to phylum Chlorophyta increased from 2.3% in the May open-sea surface water sample to 23.91% in July, whereas those in phylum Ochrophyta decreased from 36.66% in May to 7.32% in July (Table 3). The major differences between the May and July open-sea deep water samples were increases in proportions of species belonging to the Dinophyta (from 2.33% to 13.01%) and Ochrophyta (from 17.74% to 28.48%) (Fig. 2).

Table 4 Top 20 phytoplanktonic OTUs in each sample of East/Japan Sea.

M I S	M II S	M II D	J I S	J II S	J II D	
OTUs	%	OTUs	%	OTUs	%	OTUs	%	OTUs	%	OTUs	%	
Coscinodiscophyceae-PKNUE1	58.34	Cyanophyceae-PKNUE1	14.21	Heterosigma akashiwo-EU168190	7.47	Coscinodiscophyceae-PKNUE52	23.33	Mamiellophyceae-PKNUE1	7.06	Bacillariophyceae-PKNUE4	12.93	
Coscinodiscophyceae-PKNUE2	2.63	Coscinodiscophyceae-PKNUE1	11.79	Coccolithophyceae-PKNUE71	3.20	Coscinodiscophyceae-PKNUE55	16.61	Synechococcus sp.-CP000110	6.48	Dinophyceae-PKNUE10	10.89	
Coscinodiscophyceae-PKNUE79	2.56	Heterosigma akashiwo-EU168190	9.91	Fragilariophyceae-PKNUE1	1.90	Coscinodiscophyceae-PKNUE25	9.04	Dinophyceae-PKNUE10	3.84	Coccolithophyceae-PKNUE4	2.49	
Dinophyceae-PKNUE7	1.63	Synechococcus sp.-CP000097	5.04	Phaeocystis antarctica-JN117275	1.24	Coscinodiscophyceae-PKNUE56	4.67	Coccolithophyceae-PKNUE4	3.59	Mamiellophyceae-PKNUE8	1.58	
Coscinodiscophyceae-PKNUE49	0.99	Coccolithophyceae-PKNUE42	1.94	Coscinodiscophyceae-PKNUE92	0.89	Coscinodiscophyceae-PKNUE57	3.69	Micromonas pusilla-FN563097	3.37	Coscinodiscophyceae-PKNUE55	1.46	
Coscinodiscophyceae-PKNUE41	0.89	Dinophyceae-PKNUE10	1.93	Coscinodiscophyceae-PKNUE1	0.83	Coscinodiscophyceae-PKNUE58	2.18	Prochlorococcus marinus-CP000576	3.34	Coscinodiscophyceae-PKNUE74	1.34	
Coscinodiscophyceae-PKNUE4	0.87	Coccolithophyceae-PKNUE5	1.81	Coscinodiscophyceae-PKNUE51	0.73	Dinophyceae-PKNUE4	2.07	Mamiellophyceae-PKNUE3	3.04	Coscinodiscophyceae-PKNUE90	1.22	
Coscinodiscophyceae-PKNUE3	0.76	Coccolithophyceae-PKNUE6	1.08	Cryptophyceae-PKNUE1	0.69	Coscinodiscophyceae-PKNUE59	1.87	Mamiellophyceae-PKNUE6	3.02	Dinophyceae-PKNUE6	1.21	
Coscinodiscophyceae-PKNUE40	0.72	Coscinodiscophyceae-PKNUE13	1.06	Coccolithophyceae-PKNUE50	0.68	Dinophyceae-PKNUE10	1.80	Synechococcus sp.-CP006882	2.40	Coccolithophyceae-PKNUE38	1.18	
Coscinodiscophyceae-PKNUE19	0.65	Dinophyceae-PKNUE7	1.04	Dinophyceae-PKNUE11	0.64	Coscinodiscophyceae-PKNUE60	1.67	Mamiellophyceae-PKNUE2	2.18	Mamiellophyceae-PKNUE1	1.09	
Coscinodiscophyceae-PKNUE91	0.62	Coscinodiscophyceae-PKNUE2	1.02	Mamiellophyceae-PKNUE6	0.54	Coscinodiscophyceae-PKNUE26	1.29	Mamiellophyceae-PKNUE4	1.87	Coccolithophyceae-PKNUE7	1.03	
Coscinodiscophyceae-PKNUE5	0.58	Coscinodiscophyceae-PKNUE90	0.98	Mamiellophyceae-PKNUE8	0.50	Dinophyceae-PKNUE7	1.05	Mamiellophyceae-PKNUE8	1.78	Coscinodiscophyceae-PKNUE18	1.01	
Coscinodiscophyceae-PKNUE6	0.51	Coccolithophyceae-PKNUE40	0.91	Coccolithophyceae-PKNUE73	0.49	Coscinodiscophyceae-PKNUE27	0.98	Coccolithophyceae-PKNUE50	1.69	Coscinodiscophyceae-PKNUE52	0.74	
Coscinodiscophyceae-PKNUE80	0.50	Coscinodiscophyceae-PKNUE51	0.84	Coccolithophyceae-PKNUE47	0.49	Coscinodiscophyceae-PKNUE82	0.93	Dinophyceae-PKNUE12	1.62	Coscinodiscophyceae-PKNUE76	0.73	
Dinophyceae-PKNUE8	0.45	Coccolithophyceae-PKNUE43	0.82	Coccolithophyceae-PKNUE4	0.47	Coscinodiscophyceae-PKNUE28	0.83	Synechococcus sp.-CP000097	1.45	Coscinodiscophyceae-PKNUE89	0.71	
Coscinodiscophyceae-PKNUE8	0.43	Mamiellophyceae-PKNUE8	0.77	Coscinodiscophyceae-PKNUE24	0.47	Coscinodiscophyceae-PKNUE1	0.67	Coccolithophyceae-PKNUE21	1.25	Coscinodiscophyceae-PKNUE75	0.69	
Bacillariophyceae-PKNUE2	0.42	Coccolithophyceae-PKNUE1	0.76	Mamiellophyceae-PKNUE1	0.46	Coscinodiscophyceae-PKNUE45	0.66	Coscinodiscophyceae-PKNUE94	1.23	Cyanophyceae-PKNUE12	0.68	
Coscinodiscophyceae-PKNUE21	0.40	Coccolithophyceae-PKNUE44	0.72	Coscinodiscophyceae-PKNUE52	0.46	Coscinodiscophyceae-PKNUE61	0.59	Mamiellophyceae-PKNUE5	1.09	Coccolithophyceae-PKNUE64	0.66	
Coscinodiscophyceae-PKNUE48	0.40	Fragilariophyceae-PKNUE1	0.68	Coscinodiscophyceae-PKNUE20	0.43	Coscinodiscophyceae-PKNUE62	0.55	Coccolithophyceae-PKNUE54	0.84	Coccolithophyceae-PKNUE46	0.65	
Dinophyceae-PKNUE1	0.40	Coscinodiscophyceae-PKNUE92	0.68	Coccolithophyceae-PKNUE15	0.42	Coscinodiscophyceae-PKNUE63	0.54	Cyanophyceae-PKNUE5	0.69	Bacillariophyceae-PKNUE5	0.64	

Discussion

Reliability of the modified universal primers for algal community structure analysis

The modified universal primers were more effective at identifying prokaryotic and eukaryotic phytoplankton taxa, and effectively excluded proteobacterial sequences. Compared with 16S primers, the modified primers will be more useful for analyzing phytoplankton community structure, as they allow for identification of more photosynthetic phytoplankton. When data produced by both 16S and 23S primers are compared, it will be possible to assess for correlations between autotrophic phytoplankton species and heterotrophic bacteria in the ocean. In the present study, we also found that all the sequences belonging to the Chlorophyta contained a GGACAA sequence rather than GGACAG at the 5′ end of the original universal forward primer region (Table 1). This finding enabled us to modify the primers such that we were able to detect a greater proportion of Chlorophyta in the surface water of the open sea, and this supports the value of the modified universal primers (Fig. 4). In addition, the nested PCR using a low cycle number and modified universal primers yielded more eukaryotic algal sequences than the original primers did. Use of a single primer set often results in exaggerated stochastic effects due to the exponential nature of PCR amplification and variable primer efficiency (Caragine et al., 2009; Pinto & Raskin, 2012). The original 23S primers, which cannot discriminate heterotrophic proteobacteria from other photosynthetic phytoplankton species (whereas 16S primers can do so), yielded fewer eukaryotic algal sequences when the protobacterial population was large (Table 2). To overcome those drawbacks and to present better quantitative phytoplankton community structure, it is necessary to use reduced PCR cycles and different primer sets. In the present study, a nested PCR strategy with low cycle number reduced the stochastic effects of PCR and yielded more photosynthetic algal species sequences presenting the better phytoplankton community structure for each sample examined. In conclusion, the performance of the modified 23S universal primers was superior to that of the original universal primers designed for marine phytoplankton community structure analysis.

Of the 345 OTUs identified in this study, only 95 (27.5%) matched the sequences previously listed in the GenBank database, and this is an important limitation of the 23S primers. This is mainly because most phytoplankton researchers have paid more attention to the molecular taxonomy of the 16S rDNA region than the 23S rDNA region. Although many more 16S rDNA sequences have been deposited in the database, it is difficult to design the “universal primers” that will amplify all the prokaryotic and eukaryotic algal taxa that correspond to the 16S rDNA region. In fact, most studies that have used the universal primers have been conducted to determine the biodiversity of a specific taxon or group (Asudi et al., 2016; Cruaud et al., 2014; Kitamura et al., 2016; Logares et al., 2014; Valenzuela-González et al., 2016; Vierheilig et al., 2015). In contrast, our current results with the modified primers that target plastid 23S rDNA indicate that these can be used as “universal primers” for analyzing algal community structure because of their large coverage and the fact that they amplify both prokaryotic and eukaryotic algal species. For this reason, if PHYTOGEN is operated properly documenting the plastid 23S rDNA sequences this would be a useful database for understanding the marine phytoplankton community. Without doubt, the first step to improving the PHYTOGEN database would be to increase the sequence data with species description. Since a great deal of 16S rDNA data have already exist, one of the fastest ways to increase the sequence information is to perform massive sequencing the long DNA fragments that contain both 16S rDNA and 23S rDNA regions within the same sample. Given that the cost for sequencing is steadily decreasing, it is expected that the availability of plastid 23S rDNA sequences will increase dramatically and this will enable researchers to ultimately obtain extensive, useful 23S rDNA data without any further morphological analysis. In addition, the PHYTOGEN database will need to be updated periodically to add new findings for 23S rDNA to the GenBank database. Once PHYTOGEN has been updated and higher resolution data become available, we will produce data with better resolution and our dependence on molecular surveys for marine phytoplankton evaluations will increase.

Effects of economic metabarcording strategy on marine phytoplankton studies

After processing the raw reads, the numbers of end-paired amplicons obtained from each respective sample ranged from 290,082 to 639,836, and averaged 438,896 (Table 3). Compared with the previously reported average amplicon numbers produced by the Roche 454 pyrosequencing system (Eiler et al., 2013), equivalent data have been obtained at less than 1/100th of the cost using the Illumina platform (Mardis, 2008). Based on this, several studies have since used the MiSeq platform to analyze bacterial community structure using 16S primers (Caporaso et al., 2012; Schmidt et al., 2013); however, “universal” primers that are suitable for the Illumina platform (400 bp–500 bp) and cover prokaryotic and eukaryotic phytoplankton have not been reported. The modified primers we designed for this study are not only optimized for the Illumina platform with high resolving length (404–411 bp), but also exhibit large taxonomic coverage, amplifying prokaryotic as well as eukaryotic phytoplankton. In our present study, compared to the Roche 454 system, we obtained similar data at 1/20th the cost using the Illumina MiSeq system after trimming the raw data with high quality (6 bp overlapping sequences and omitting any mismatches). The cost would be further reduced to reach a similar conclusion by the increased indices or the less strict trimming algorithm. Considering the low cost of the metabarcoding strategy, it is possible for a single laboratory to run a year-round phytoplankton community survey handling the several thousand samples, which has never been imagined before. This means that researchers should spend more time in the laboratory analyzing the huge amount of data using various bioinformatics tools instead of collecting samples in the field.

This economic metabarcoding strategy enables large amounts of data to be analyzed at one time and will allow researchers to obtain novel knowledge. We recognize that certain features of the sampling methods should be changed; specifically, the filtered water volume should be 10 L per sample, as opposed to the 0.5 L volume used in our phytoplankton survey. A previous study showed that OTU richness was greater with increased filtered water volume (Padilla et al., 2015); therefore, increasing the water volume per sample would generate more accurate results, less variability and higher OTU numbers. If the proportion of each OTU is linked to the total carbon values, we would also be able to estimate the carbon biomass of each OTU. We could construct a spatio-temporal database of phytoplankton community changes on a daily or even hourly basis. Most of all, this new strategy would be especially useful for analyzing phytoplankton community structure and its contribution to primary production in ocean. One long-term question is whether picoplankton are expected to gradually increase as the oceans become warmer (Morán et al., 2010). These data would provide novel information about the paradoxical nature of phytoplankton (Hutchinson, 1961) and its ecosystem functions in marine food webs, which have not been investigated to date.

Supplemental Information

Supplemental Information 1 Supplemental hTable

Click here for additional data file.

Additional Information and Declarations

Competing Interests

Author Contributions

Field Study Permissions

Data Availability

The authors declare there are no competing interests.

Tae-Ho Yoon conceived and designed the experiments, performed the experiments, wrote the paper, prepared figures and/or tables.

Hye-Eun Kang conceived and designed the experiments, performed the experiments, prepared figures and/or tables.

Chang-Keun Kang analyzed the data, contributed reagents/materials/analysis tools.

Sang Heon Lee analyzed the data.

Do-Hwan Ahn performed the experiments.

Hyun Park performed the experiments, wrote the paper, reviewed drafts of the paper.

Hyun-Woo Kim conceived and designed the experiments, contributed reagents/materials/analysis tools, wrote the paper, reviewed drafts of the paper.

The following information was supplied relating to field study approvals (i.e., approving body and any reference numbers):

1. National Fisheries Research & Development Institute (NFRDI, current name is National Instititute of Fisheries Sciences(NIFS)).

2. Department of Fisheries Resource Management-1728 (2014-05.22) & Department of Fisheries Resource Management-2506 (2014-05.22).

The following information was supplied regarding data availability:

Figshare: https://figshare.com/articles/JIID_R1_fastq_gz/2131864.

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
