# Peer review of "Development of a cost-effective metabarcoding strategy for analysis of the marine phytoplankton community"

_PeerJ, doi:10.7717/peerj.2115_

## Round 0.1 · original submission · Major Revisions

To resubmit your manuscript it is critical to address each comment raised by the two reviewers. Importantly, the manuscript will need to be proofread and edited for language/ grammar by an English expert. Further, the current lack of references for tag sequencing of phytoplankton will need to be addressed (see reviewer #2) and the manuscript text modified accordingly to credit these previous studies.

Currently, the database used in this study is not publicly available, which will preclude publication. Please make the database publicly available to the reviewers.

Please address the criticism raised in regards to the display items (Figures and Table1) and all additional minor points raised by the reviewers.

Reviewer 1 ·

Basic reporting

The article has many sentences with grammar related issues (lines 69, 150, 173, 196, 268). I would recommend that the authors send this manuscript to an English language expert or spend more time to examine these issues. The introduction is sufficiently broad that it explains how the work fits within the broader field of knowledge. The structure of the article appears to fit one of the standard templates. There are issues with several of the figures:

Figure 1. The legend does not pertain to the Figure in question. This legend should instead be included in the materials and methods, or a new figure should be generated.

Figure 2. This figure would benefit from greater detail; it is unclear what the "update" process is referring to.

Figure 3. The use of pie charts is inappropriate; not only are there too many categories (many of which are not shown in the pie chart) listed for the reader to keep track of, some of the categories use the same colors. Stacked barcharts with fewer categories is a better way to achieve this. Figure 3A is challenging, but Figure 3B is impossible to interpret.

Figure 4. More attention should be paid to how the group names are being displayed. Perhaps this tree could be condensed in some way so as to be more readable.

Figure 5A. See comments from Figure 3 above. Figure 5B. The names used to describe the surface samples are not the same names used throughout the manuscript. This must be corrected.

Table1. There is no need to provide the full OTU table as a supplemental table within the Manuscript. This is simply too much to display in tabular format. Please instead reference an archived OTU table in BIOM format.

The resolution of several of the figures is below publication standard and should be increased (in particular, Figure 3 and 5).

Experimental design

The research does describe original research. The submission does define the research question, which is to develop novel method for increased cost-effectivness for phytoplankton community profiling. The design of the nested primers and amplification strategies appear technically sound. The methods are described in sufficient detail.

Validity of the findings

The data is from an experiment that by design prevented the statistical assessment of variation between replicates. For this reason, it is impossible to assert with any real confidence that the trends observed between sites and depths are representative of these environments. The study does however offer a novel barcoding design for phytoplankton research, which by itself is likely to be of benefit to their specific research community.

Additional comments

I have several general comments to the author.

1) I do not understand the use of the term MOTU, instead of the more conventional "OTU".
2) I do not understand the use of the term "contig" instead of the more conventional term "amplicon".
3) It might be useful to describe the calculation that the authors have done to indicate the reduction in cost (1/20th of standard costs).
4) I am confused by the assertion in line 152 that you have identified 107 MOTUs, which is followed by the line 154 which states that you have identified 73 MOTUs.
5) I did not follow the explanation on lines 236-238 about "suppression" of eukaryotic signal. This should be reworded or reworked.
6) The description of the PHYTOGEN database contsruction was a little unclear; it seems that an iterative process of comparing 16S amplicons from this project to existing reference sequences, followed by adding novel amplicons as new references, was employed. Is this correct?

Reviewer 2 ·

Basic reporting

See below for details but in brief
1. The authors fail to cite the numerous papers that have been publised using HTS for phytoplankton
2. The English really needs to be improved

Experimental design

More information need to be provided on primer design. Database structure must be detailed and database should be made public.

Validity of the findings

Data are OK but not discussed at all in context of existing data.

Additional comments

General remarks
This paper introduces a new approach to obtain information concerning the composition of phytoplankton populations using the 23S of the plastid rRNA gene. Previous approaches include the use of V4 and V9 regions of the 18S and the 16S platid gene. The disadvantage of the approach chosen is that the plastid 23S reference database is very limited (ie. we have very few 23S sequences for known marine taxa) while its advantage could be a better resolution than the 16S.
My main concern with the paper is the total absence of references on the use of metabarcoding (tag sequencing) to characterize phytoplankton. There is a plethora of references using 18S V4 and V9 and none appears in the paper. The authors preent their method as completely new, while for sure they should have discussed the advantage of their approach and their results in light of exisiting litterature. Their absence of knowledge of the litterature leads also to very naive statements.
Another point is whether the PHYTOGEN database they build is building. For me a prerequisite of the publication of the paper is to make the database freely available.
Besides there are numerous points that require clarification and the english will need to be improved.



Specific remarks
l. 44 obvious
l. 56 Cite more recent systems which have much better resolution such as Flow CytoBot : Olson, R.J., Shalapyonok, A. & Sosik, H.M. 2003. An automated submersible flow cytometer for analyzing pico- and nanophytoplankton: FlowCytobot. Deep - Sea Res. Part I - Oceanogr. Res. Pap. 50:301–15. Campbell, L., Olson, R.J., Sosik, H.M., Abraham, A., Henrichs, D.W., Hyatt, C.J. & Buskey, E.J. 2010. FIRST HARMFUL DINOPHYSIS (DINOPHYCEAE, DINOPHYSIALES) BLOOM IN THE U.S. IS REVEALED BY AUTOMATED IMAGING FLOW CYTOMETRY. J. Phycol. 46:66–75.
l. 65. What do this number of plastid sequences correspond to? Is this all genes from plastids ? If the authors refer to plastid 16S rRNA the number is much smaller. See : Decelle, J., Romac, S., Stern, R.F., Bendif, E.M., Zingone, A., Audic, S., Guiry, M.D. et al. 2015. PhytoREF: a reference database of the plastidial 16S rRNA gene of photosynthetic eukaryotes with curated taxonomy. Mol. Ecol. Resour. 15:1435–45. More generally the authors have overlooked this paper which also deals with plastid rRNA metabarcodes.
l. 77 Working length for 454 is more like 500 bp. 700 bp. is maximum but was rarely achieved.
l. 79. The author should indicate that the 454 is now phased out and not sold anymore...
l. 79-84. The authors have completely overlooked the recent litterature using NGS to determine phytoplankton (and protist). Here are a few references that they should mention (but many more have come out in the last 2 years)
Egge, E., Bittner, L., Andersen, T., Audic, S., de Vargas, C. & Edvardsen, B. 2013. 454 Pyrosequencing to Describe Microbial Eukaryotic Community Composition, Diversity and Relative Abundance: A Test for Marine Haptophytes. PLoS One. 8:e74371.
Kilias, E., Wolf, C., Nöthig, E.-M., Peeken, I. & Metfies, K. 2013. Protist distribution in the Western Fram Strait in summer 2010 based on 454-pyrosequencing of 18S rDNA. J. Phycol. 49:996–1010.
Egge, E.S., Johannessen, T.V., Andersen, T., Eikrem, W., Bittner, L., Larsen, A., Sandaa, R.-A. et al. 2015. Seasonal diversity and dynamics of haptophytes in the Skagerrak, Norway, explored by high-throughput sequencing. Mol. Ecol. in press.
Boopathi, T., Lee, J.-B., Youn, S.H. & Ki, J.-S. 2015. Temporal and spatial dynamics of phytoplankton diversity in the East China Sea near Jeju Island (Korea): A pyrosequencing-based study. Biochem. Syst. Ecol. 63:143–52.
Metfies, K., von Appen, W.-J., Kilias, E., Nicolaus, A. & Nöthig, E.-M. 2016. Biogeography and Photosynthetic Biomass of Arctic Marine Pico-Eukaroytes during Summer of the Record Sea Ice Minimum 2012. PLoS One. 11:20 pp.
Le Bescot, N., Mahé, F., Audic, S., Dimier, C., Garet, M.-J., Poulain, J., Wincker, P. et al. 2015. Global patterns of pelagic dinoflagellate diversity across protist size classes unveiled by metabarcoding. Environ. Microbiol. n/a – n/a.
de Vargas, C., Audic, S., Henry, N., Decelle, J., Mahe, F., Logares, R., Lara, E. et al. 2015. Eukaryotic plankton diversity in the sunlit ocean. Science. 348:1261605.
Massana, R., Gobet, A., Audic, S., Bass, D., Bittner, L., Boutte, C., Chambouvet, A. et al. 2015. Marine protist diversity in European coastal waters and sediments as revealed by high-throughput sequencing. Environ. Microbiol. 17:4035–49.
l. 113 and L116. Show location of primers on sequence in supplementary figure. Also it would be good to show an alignement with representative of all major classes of phuytoplankton.
l. 137. The strucuture of the PHYTOGEN database should be explained better in comparison in particular of the SILVA, PR2 and PHYTOREF databases. For example, how many taxonomic levels are used. Also the database should be made public.
l. 137. Were all 23S plastid sequences available in GenBank extracted. What is the taxonomic distributions of the PHYTOGEN database
l. 139. There is no reason to exclude teh uncultured sequecnes from teh database. A careful phylogenetic analysis should allow to obtain some informaton n their taxonomy. Again see the PR2 and PhytoRef databases for which "uncultured" sequences have been anotated.
l. 142. Why annotate only at the class elvel. In some cases you maybe able to go down to genus or event species.
l. 144 and elsewhere. There is aboslutely no need to create a new abbreviation. Just use OTU as everyone else.
l. 150. Not clear what is compared... Is it one round of amplification vs. two rounds.
l. 163. What do you mean by suppresion effects. Please provide references.
l. 170 Our curent knowledge of plastid 23S barcode is very limited because there are very sequences available but it is not true of phytoplankton. Again please read about the PR2 and PhytoRef databases.
l. 173. Again a very vague and completely wrong statement. We know quite a lot about algal taxonomy and papers are published everyday about this field.
l. 179. What do you mean ? The sentence construction is just wrong. You do not use a % similairity to amplify a sequence.
l. 183 "not identified", you mean not found ?
l. 183. Since the primers target the plastid and therefore autotrophs it is logical that you do not find taxa that are heterotrophic
l. 195. Since your target photosynthetic groups and the bacteria are contaminants due to no specificity of primers, you should remove from your further analysis all bacterial sequences.
l. 198. Another neologism "Uni-MOTU", please stick to established terminology and do not make your own...
l. 209. Again, the authors have no knowledge of the relevant litterature. There is a huge litterature of marine cyanobacteria distribution in the ocean and on the relevant parameters... Please read and cite
l. 216-218. Again please read and cite litterature
l. 233-234. Again I do not understand this statement about exponential nature of PCR amplification.
l. 239. Stochastics effect can be reduced by other means such as PCR pooling.
l. 242. Which primer, forward or reverse ?
l; 250. Many other studies with MiSeq, so nothing new here....
l. 255. This has been done already see for example the Tara papers.
l. 258-260. No this is jsut due to the fact that nobody has used before the marker you are using...
l. 265. Not true... Identification to the genus or even species level is possible with 18S V4 or V9 and even 16S plastid. So this is probably the same for 23S.
l. 270. How do you know that these sequences correspond to zoospores of Rhodophyta, could be broken leaves. You identify here the species while in the previous paragraph you just said that identification at the species level was not possible with 23S (which I disagree with but sill you contradict yourself). Moreover, this is very anecdotal. Please discuss your data in the context of the existing litterature on phytoplankton...
l. 275. The black pine could be a contamination. Most studies dealing with phytoplankton discard these streptophyta sequences that are considered as contamination.
l. 283. Some studies have looked at the effect of the filtered volume. See Padilla, C.C., Ganesh, S., Gantt, S., Huhman, A., Parris, D.J., Sarode, N. & Stewart, F.J. 2015. Standard filtration practices may significantly distort planktonic microbial diversity estimates. Front. Microbiol. 6:1–10.
Fig. 3. Betaproteobacteria appear twice in Fig. 3B
Fig. 4 is impossible to read, provide as accessory data.
Fig. 5. What is the difference with Fig. 3. Which cutoff is used ?

---

## Round 0.2 · accepted · Accept

The manuscript has greatly improved and all major and minor concerns of both reviewers were appropriately addressed.